# Rapid Identification of Kudzu Powder of Different Origins Using Laser-Induced Breakdown Spectroscopy

**DOI:** 10.3390/s19061453

**Published:** 2019-03-25

**Authors:** Fei Liu, Wei Wang, Tingting Shen, Jiyu Peng, Wenwen Kong

**Affiliations:** 1College of Biosystems Engineering and Food Science, Zhejiang University, 866 Yuhangtang Road, Hangzhou 310058, China; fliu@zju.edu.cn (F.L.); 15236193955@163.com (W.W.); shentingtingstt@163.com (T.S.); jypeng@zju.edu.cn (J.P.); 2School of Information Engineering, Zhejiang A & F University, 666 Wusu Street, Hangzhou 311300, China

**Keywords:** kudzu powder, laser-induced breakdown spectroscopy, rapid identification, discrimination model

## Abstract

The rapid identification of kudzu powder of different origins is of great significance for studying the authenticity identification of Chinese medicine. The feasibility of rapidly identifying kudzu powder origin was investigated based on laser-induced breakdown spectroscopy (LIBS) technology combined with chemometrics methods. The discriminant models based on the full spectrum include extreme learning machine (ELM), soft independent modeling of class analogy (SIMCA), K-nearest neighbor (KNN) and random forest (RF), and the accuracy of models was more than 99.00%. The prediction results of KNN and RF models were best: the accuracy of calibration and prediction sets of kudzu powder from different producing areas both reached 100%. The characteristic wavelengths were selected using principal component analysis (PCA) loadings. The accuracy of calibration set and the prediction set of discrimination models, based on characteristic wavelengths, is all higher than 98.00%. Random forest and KNN have the same excellent identification results, and the accuracy of calibration and prediction sets of kudzu powder from different producing areas reached 100%. Compared with the full spectrum discriminant analysis model, the discriminant analysis model based on the characteristic wavelength had almost the same discriminant effects, and the input variables were reduced by 99.92%. The results of this research show that the characteristic wavelength can be used instead of the LIBS full spectrum to quickly identify kudzu powder from different producing areas, which had the advantages of reducing input, simplifying the model, increasing the speed and improving the model effect. Therefore, LIBS technology is an effective method for rapid identification of kudzu powder from different habitats. This study provides a basis for LIBS to be applied in the genuineness and authenticity identification of Chinese medicine.

## 1. Introduction

*Pueraria lobate* (Willd.) Ohwi is a perennial herbaceous vine of genus Leguminosae, and its root is a kind of medicinal and edible plant, also named kudzu [1]. As a famous traditional Chinese herb with a long history, kudzu has extremely high nutritive value and good medicinal effects [2]. There are many nutritional ingredients in kudzu, such as amylum, dietary fiber, protein, crude fat and flavone, which contain a variety of essential amino acids and mineral elements [3,4,5]. The main active ingredients of kudzu are isoflavones, which have many pharmacological effects [3,6]. Kudzu contains abundant isoflavones: it has been proved to be one of the plants with the highest content of isoflavones in natural plants [7]. The most valuable isoflavones in kudzu are daidzein, which can effectively relieve the symptoms of patients with alcohol poisoning, and pueraria, which plays an important role in the treatment of cardiovascular and cerebrovascular diseases [8,9,10]. In addition, pueraria also contains the ingredients of pueraria saponins and is a derivative of dihydrochalone, which can effectively protect human liver function [11]. Recent research indicates that kudzu has such pharmacological effects as anti-cardio-cerebrovascular disease [12], reducing blood sugar [13], enhancing immunity [14], relieving diarrhea and improving osteoporosis [15]. Tam et al. presented their investigation into kudzu’s effects to treat angina and other heart symptoms and found that kudzu can decrease LDL cholesterol [16]. Lee et al. showed that kudzu could help alleviate the adverse effects of ethanol ingestion [17]. Su et al. found that kudzu extract was effective for inhibition of cytopathy [1]. 

Kudzu is abundant in the mountainous areas of China, mainly distributed in the provinces of Zhejiang, Hubei, Hunan, Yunnan and Anhui. In general, different regions have different ecological environments in terms of soil, temperature, water, light and altitude. The same kind of medicinal materials from different origins will have different effects due to differences in climate and geographical location [2,18,19,20]. Therefore, origin is the main factor affecting the quality of medicinal materials. The environmental conditions directly affect the quality of samples, including active ingredients (mineral nutrition element), appearance traits (color and size), and hazardous substances (pesticide residues and heavy metals). For kudzu from different habitats, the contents of total flavonoids, puerarin and daidzein are quite different. 

With the improvement of people’s living standards, consumer demand for high-quality natural medicines like kudzu is increasing. Lured by the temptation of economic profits, some merchants have replaced the high-quality kudzu in expensive areas with poor-quality kudzu in cheap areas. In order to protect the legitimate rights and interests of consumers, and put an end to such unfair competition, a rapid and effective method should be adopted to realize the origin discrimination of kudzu. Therefore, identifying the genuineness of kudzu is of great significance for guaranteeing its authenticity and safety [18].

Laser-induced breakdown spectroscopy (LIBS) is an atomic emission spectroscopy technique that has emerged in recent years [21,22,23]. A high energy pulsed laser is used to heat and ablate the sample [24]. Then, plasma is created above the surface of the sample. The spectral signals emitted by plasma can be obtained and recorded for qualitative and quantitative analyses of the samples [24,25,26]. Currently, the methods used for detection of traditional Chinese medicine mainly include high performance liquid chromatography (HPLC), atomic absorption spectroscopy (AAS), atomic fluorescence spectroscopy (AFS), and inductively coupled plasma mass spectrometry (ICP-MS). In general, the preparation of samples, pre-processing, and laboratory chemical analysis are needed during traditional detection, and they are cumbersome, costly and time-consuming. Compared with traditional detection methods, LIBS technique has the advantages of fast analysis speed, little sample preparation, and simultaneous analysis of multiple elements [21,25]. By analyzing the spectral signal emitted from laser plasma, LIBS has been widely used in the quantitative and qualitative analysis of plant materials [21,25,27,28,29], animal tissue [24,30], mineral resources [31,32], industrial application [33] and so on. The previous studies provide a basis for LIBS to be applied in the authenticity identification of Chinese medicine.

In recent years, LIBS technique, combined with chemometrics methods, has gradually been applied to qualitative and quantitative analysis of Chinese herbal medicine. Wang et al. presented their investigation on Chinese herbal medicines. The roots of *Angelica sinensis*, *Codonopsis pilosula*, and *Ligusticum wallichii* were analyzed and identified using LIBS. The results show that LIBS combined with PCA and BP-ANN is a useful tool for identification of Chinese herbal medicine [34]. Wang et al. also used LIBS to determine copper and lead in *Ligusticum wallichii*. The results demonstrate that multiple linear regression coupled with LIBS is suitable for the determination of heavy metals in Chinese traditional medicine [35]. Hu et al. have determined the geographical origin of kudzu root based on Fourier-transform infrared spectroscopy (FT-IR) and chemometric analysis. The high recognition capability of the PLS-DA model proved that it was suitable for tracing the geographic origin of kudzu root and quantifying adulterants in kudzu root products [36]. However, there are few reports on the identification of Chinese medicinal materials from various areas using LIBS [34]. In particular, the identification of Kudzu from different origins based on LIBS technology is largely unknown. Based on LIBS technology combined with chemometrics methods, this paper established a discriminant model to study the feasibility of rapidly identifying kudzu powder from different producing areas.

## 2. Materials and Methods

### 2.1. Experimental Setup

The LIBS system used in the experiment is shown in Figure 1. Based on previous research, Q-switched Nd:YAG pulsed laser (Vlite-200, Beamtech Optronics, Beijing, China) was used to generate a pulse with a maximal energy of 200 mJ @532 nm, with a pulse duration of 8 ns. The laser was emitted and passed through the optical system. Finally, under the action of a plano-convex lens (f = 100 mm), the laser was focused 2 mm below the sample and formed a spot with a diameter of 7 mm. The echelle spectrograph (ME5000, Andor, Belfast, UK) was used to split signals generated during the transition of plasma. The spectra between 200 and 975 nm with high resolution (λ/Δλ = 5000) was collected. The optical signals were converted into electrical signals by iStar DH334 ICCD detector (DH334-18F-03, Andor, Belfast, UK), and read and recorded by computer. The digital delay generator (DG645, Stanford Research Systems, Sunnyvale, CA, USA) was used for timing control of lasers and ICCD detectors. The X-Y-Z motorized stage (TSA50-C, Zolix, Beijing, China) and the stage controller (SC300-3A, Zolix, Beijing, China) were used to move the sample. The energy meter (StarLite, Ophir, Jerusalem, Israel) consists of a probe and a meter, which were mainly used to monitor the energy of the laser pulse. In this work, the parameters of laser pulse wavelength, laser pulse energy, spectrometer delay time and ICCD detector gate width affect the experimental results. In order to increase the signal-to-noise ratio, the following parameters are optimized: λ = 532 nm, Energy = 60 mJ, Delay = 1.5 μs, Gate width = 10 μs.

### 2.2. Experimental Materials

Different kinds of pure kudzu powder from five different provinces, which are the main regions of kudzu powder production in China, were purchased from a supermarket and selected as experimental samples. The brands of pure kudzu powder used included Jiahui from Zhejiang, Guosenyuan from Hubei, Gexishi from Hunan, Xuanqing from Yunnan, and Tashantashui from Anhui. Before LIBS measurements, the samples were pressed into tablets. In the experiment, about 0.5 g powder was placed in the tableting machine (SCJS, Tianjin, China) and pressed at a pressure of 20 MPa for 1 min to prepare a tablet with a diameter of about 15 mm and a thickness of about 2 mm. The whole sample pretreatment for acquisition of LIBS signals was less than 2 min including weighing and pressing. Finally, 50 tablets were prepared for each variety of kudzu powder. Therefore, a total of 250 tablets were prepared. In order to establish classification models and verify their performance, the samples for each variety were randomly divided into calibration and prediction sets at a 3:2 ratio. As a result, the 250-sample dataset was divided into 150 samples for calibration and 100 samples for validation. The samples in the calibration set were used to establish discrimination models and optimize the model parameters, and the model accuracy was evaluated with the samples in the validation set.

### 2.3. Data Acquisition

Considering the impacts caused by internal component differences and surface impurities of samples, the following methods were used for data acquisition. In order to avoid continuous ablation of the same spot, 4 × 4 array was set as the ablation path by X-Y-Z motorized stage, where the distance between each row and each column is 2 mm. As a result, spectral data acquisition of the sample was performed on 16 locations, which are evenly distributed on each sample. To reduce fluctuation between the laser point-to-point and get stable signals, each point was cumulatively acquired 5 times with a repetition rate of 1 Hz. Finally, the average of the 80 spectra (4 × 4 × 5) was recorded as the LIBS data of the sample. A total of 250 LIBS spectral data representing 250 samples of kudzu powder were obtained in the experiment. The time of LIBS information collection for one sample was about 1 min. And it is compatible with the requirements of on-site analysis.

### 2.4. Selection of Characteristic Wavelength

LIBS technology can simultaneously collect multiple element spectral lines and large amounts of data, including a lot of unnecessary information such as redundancy, collinearity and background information. Therefore, models based on full-spectrum data are prone to problems such as complex models, large amounts of computation and long calculation times, which affect the extraction of useful information and are not conducive to the rapid identification of kudzu powder in different regions. The selection of characteristic wavelengths, by selecting wavelengths containing useful information and eliminating wavelengths representing redundancy, collinearity and background signals from full-spectrum data, achieves purposes such as reducing input, simplifying models, increasing speed and improving model effects [37,38].

In this study, principal component analysis (PCA) loadings was used to select characteristic wavelengths. Principal component analysis is a multivariate statistical method commonly used for material classification [34,39]. It reconstructs data through a series of linear combinations to achieve the purpose of classifying samples [34]. Principal component analysis loadings is a new variable related to the wavelength variable obtained in PCA, which reflects the degree of correlation between the principal component and the original wavelength of the spectrum. In the process of PCA, every principal component will give the PCA loadings a diagram of all wavelengths. There is a positive correlation between the absolute value of load and the extent to which its wavelength affects the prediction ability of the model. Therefore, we should select the wavelengths in the principal components, which have the larger absolute value of load, as the characteristic wavelengths. In the experiment, five kinds of kudzu powder from different provinces were analyzed by PCA. The number of principal components was determined according to the contribution rate of a single principal component, and the peaks and troughs were selected as the characteristic wavelengths based on the loadings plots of a principal component.

### 2.5. Discriminant Analysis Method

Extreme learning machine (ELM) is a single-hidden layer feedforward neural network (SLFNN) [40,41]. When running ELM, only one parameter of the number of hidden layer nodes needs to be set and the global optimal solution is obtained by outputting the weight. Extreme learning machine has the advantages of fast learning speed, strong generalization ability and the ability to generate a unique optimal solution. Therefore, ELM algorithm can be used for discriminant analysis [42,43].

Random forest (RF) is composed of multiple decision trees to form a decision forest. The combination of Bagging (boostrap aggregating) algorithm and Randomization algorithm construct a decision device, which uses output of different decision trees as output results. Random forest can also be used for classification and regression [44].

K-nearest neighbor (KNN) calculates the distance between the sample and the remaining samples, selects k samples from the nearest distance, then classifies the samples into the category with the largest proportion, and finally achieves the classification of the samples [30].

The soft independent modeling of class analogy (SIMCA) is a supervised pattern recognition method based on PCA [45]. The main steps of the SIMCA method are as follows: first, the principal component analysis model is established and the number of optimal principal components is selected based on the spectral data of each type of sample; then, the corresponding principal component and the principal component spectral residuals are calculated; and finally, discriminant analysis of samples is realized [46].

## 3. Results and Discussion

### 3.1. Spectral Analysis of Different Samples

The original spectra of different kudzu powders were acquired and prepared for analysis, as shown in Figure 2. In Figure 2, the abscissa is the wavelength, which ranges from 230 to 880 nm, and the ordinate is the intensity of spectra. Based on the National Institute of Standards and Technology (NIST) Atomic Spectra Database (ASD), some strong emission lines were observed in all five kinds of kudzu powder. Molecular bands CN 0-0 (around 388 nm), which usually associate with organic compounds, were observed in all five kinds of samples [47]. In addition, some atomic emission lines of C I (247.86 nm), Si I (251.61 nm), Mg II (279.55 nm, 280.27 nm), Ca II (317.93 nm, 393.37 nm, 396.85 nm, 854.21 nm), Ca I (422.67 nm), H I (656.28 nm), O I (777.54 nm), Fe I (821.58 nm), Fe II (844.80 nm) and N I (868.02 nm) were easily recognized.

Although the origin is different, the five kinds of kudzu are the same in terms of species. Their active ingredients and appearance traits are similar. As a result, the varieties of elements contained in the samples are basically the same. It is reflected in the similarity of the average spectra of the five samples. We noted that the main element peaks of different kudzu powder were similar. However, there were great differences between the intensity of spectra. The highest peak intensities of kudzu powder in Zhejiang, Hubei, Anhui, Hunan and Yunnan reached 6.2 × 10^5^, 4.5 × 10^5^, 5.4 × 10^5^, 5.1 × 10^5^ and 2.0 × 10^6^, respectively. Previous research has shown that there was a linear relationship between the peak intensity of the element in the laser-induced breakdown spectrum and the element content. As shown in the Figure 2, the content of the same elements in the kudzu powder varied among all samples. In particular, we could clearly observe that the element peak intensity of kudzu powder from Yunnan was significantly different from the other four varieties. It was found that the intensity of Mg II (279.55 nm, 280.27 nm), Ca II (393.37 nm, 396.85 nm, 854.21 nm) and Ca I (422.67 nm) of kudzu powder from Yunnan is much stronger than 7.0 × 10^5^. However, the peak of these elements in the other four samples was much lower. Therefore, we could conclude that the element contents of kudzu powder from Yunnan differ greatly from the other four. Based on this, we can quickly identify kudzu powder from different regions by spectral data processing and selection of suitable discriminant models.

### 3.2. Principal Component Analysis

Principal component analysis was performed based on the LIBS spectra of samples [31]. The scores scatter plots of PC1, PC2 and PC3 of different samples are shown in Figure 3. Each variety is represented by a different color for better visualization. Among them, PC1, PC2 and PC3 accounted for 98.41%, 0.73% and 0.40% of the total variance, respectively, whereas the first three principal components accounted for 99.55% of the total variance.

It can be seen from Figure 3 that there are five distinct clusters, which represent five different kinds of samples, in scoring spaces including PC1-PC2 space, PC2-PC3 space and PC1-PC2-PC3 space. In the PC1-PC2 space, Zhejiang and Yunnan can be completely separated from the rest, whereas groups belonging to individuals Hubei, Anhui and Hunan are hard to distinguish due to overlapping. In the space of PC2-PC3, Zhejiang was almost completely separated from the other four samples, but there was overlap between the other four samples. In the space of PC1-PC2-PC3, the five samples had a good clustering effect. The kudzu powder of the same origin was more likely to gather together.

### 3.3. Discriminant Analysis Model Based on Full Spectrum

The discriminant analysis models, including ELM, SIMCA, KNN and RF, were established based on LIBS full spectrum data. For the discriminant models, optimal parameters and the results are shown in Table 1.

It can be seen from Table 1 that the discrimination results of four models, which were based on full spectrum, were acceptable. The prediction results of ELM and SIMCA models were similar, the accuracy of the calibration set was 100%, and the accuracy of the prediction sets was 99.30% and 99.00%, respectively. Random forest and KNN had the same excellent identification results and the accuracy of calibration and prediction sets of kudzu powder from different producing areas reached 100%. The results showed that LIBS spectroscopy is feasible for rapid identification of kudzu powder from different producing areas.

### 3.4. Selection of Characteristic Spectral Lines

Generally, the amount of data in the LIBS spectrum is large. The LIBS spectrum collected in this study contained 20,937 spectral data points. The discriminant analysis model established by the full spectrum requires a large amount of calculation and a long calculation time, which not only requires high performance of the computer but also easily causes instability of the model.

It was known from the results of PCA that the first 3 principal components could explain 99.55% of the total spectral data. Therefore, we selected the characteristic variables for modeling based on the loading plots of PC1, PC2 and PC3. In the loading plots of the principal component, the larger the absolute value of the loading values corresponding to the variables, which are at the positive or negative peak, the higher the importance to the full spectrum data. In other words, such variables can retain more raw data. The loading plots of PC1, PC2, and PC3 based on full variables are shown in Figure 4.

In Figure 4, the loading values of variables at the positive or negative peaks had larger absolute values and were more important to the full data. When processing the data, we used 0.05 as the reference value and selected the variable, whose absolute value of the loading was greater than 0.05, as the alternative key variable of the principal component. Using the above method, alternative key variables for PC1, PC2 and PC3 could be selected according to Figure 4. Then, the intensity of the spectral line and the distance between the spectral peak and other surrounding peaks of the above alternative key variables were comprehensively compared. The 18 spectral lines at the peak or trough were selected as the key variables in Figure 4, based on the above selection conditions of characteristic spectral lines. According to the National Institute of Standards and Technology (NIST) Atomic Spectra Database (ASD) and relevant references, the elements corresponding to the selected 18 characteristic lines were determined as shown in Table 2.

### 3.5. Discriminant Analysis Based on Characteristic Wavelength

The discriminant analysis models of ELM, SIMCA, KNN and RF were established based on the selected characteristic wavelengths. For the discriminant models, optimal parameters and the results are shown in Table 3.

It can be seen from Table 3 that the discrimination results of four models, which were based on characteristic wavelength, were good. The prediction results of ELM and SIMCA models was similar, the accuracy of calibration set was 100%, and the accuracy of prediction sets was 99.30% and 98.00%, respectively. Random forest and KNN had the same excellent identification results and the accuracy of calibration and prediction sets of kudzu powder from different producing areas reached 100%. 

Compared with the discriminant analysis models based on full spectrum, the identification effect of the SIMCA model was reduced—the accuracy of the prediction set was reduced from 99.00% to 98.00%—whereas the other three models had the same identification results. The results showed that accuracy of discriminant models based on the characteristic wavelengths, including ELM, SIMCA, KNN and RF, was more than 98.00%. Compared with the full spectrum discriminant analysis model, the discriminant analysis model based on the characteristic wavelength had almost the same discriminant effects and the input variables were reduced by 99.92%. Therefore, the characteristic wavelength can be used instead of the LIBS full spectrum to quickly identify kudzu powder from different producing areas, which have the advantages of reducing input, simplifying the model, increasing the speed and improving the model effect.

## 4. Discussion

Based on LIBS technology combined with chemometrics methods, this paper established a discriminant model to study the feasibility of rapidly identifying kudzu powder from different producing areas.
The original spectra of different samples were acquired by LIBS. Principal component analysis was performed and the first 3 principal components had a total variance of 99.55%.The accuracy of discriminant models based on the full spectrum, including ELM, SIMCA, KNN and RF, was more than 99.00%. The prediction results of KNN and RF models were best, of which the accuracy of calibration and prediction sets of kudzu powder from different producing areas both reached 100%. The results showed that LIBS spectroscopy is feasible for rapid identification of kudzu powder from different producing areas.The accuracy of discriminant models based on 18 characteristic wavelengths, including ELM, SIMCA, KNN and RF, was more than 98.00%. Random forest and KNN have the same excellent identification results, and the accuracy of calibration and prediction sets of kudzu powder from different producing areas reached 100%. Compared with the full spectrum discriminant analysis model, the discriminant analysis model based on the characteristic wavelength had almost the same discriminant effects, and the input variables were reduced by 99.92%. The results of the research showed that the characteristic wavelength could be used instead of the LIBS full spectrum to quickly identify kudzu powder from different producing areas, which had the advantages of reducing input, simplifying the model, increasing the speed and improving the model effect.

Therefore, LIBS technology is an effective method for rapid identification of kudzu powder from different habitats. This study provides a basis for LIBS to be applied in the authenticity identification of Chinese medicine.

## Figures and Tables

**Figure 1 sensors-19-01453-f001:**
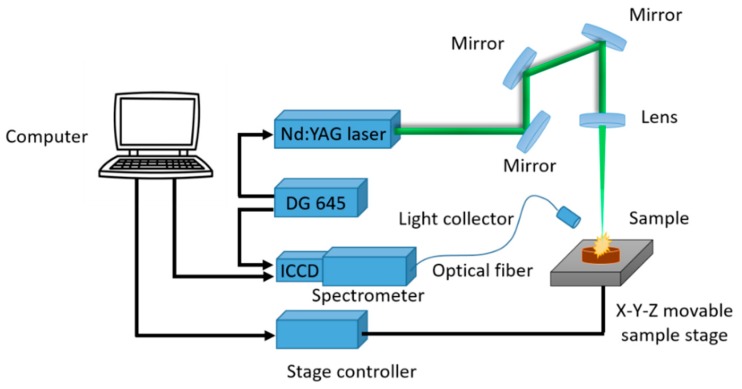
The device of laser-induced breakdown spectroscopy (LIBS).

**Figure 2 sensors-19-01453-f002:**
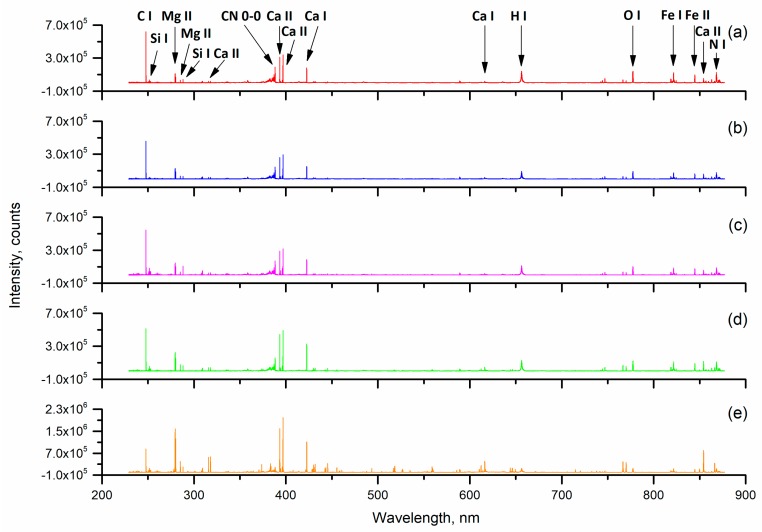
Average original spectra of kudzu powder samples. (**a**) Zhejiang; (**b**) Hubei; (**c**) Anhui; (**d**) Hunan; (**e**) Yunnan.

**Figure 3 sensors-19-01453-f003:**
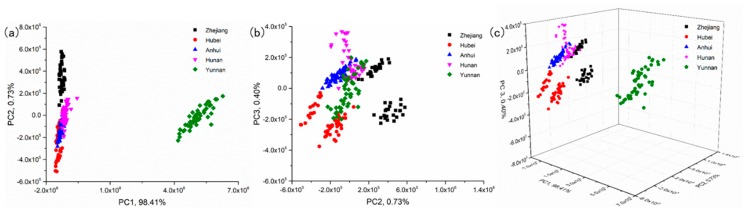
Scores scatter plots of PC1, PC2 and PC3 of different samples. (**a**) Plots of PC1-PC2; (**b**) Plots of PC2-PC3; (**c**) Plots of PC1-PC2-PC3.

**Figure 4 sensors-19-01453-f004:**
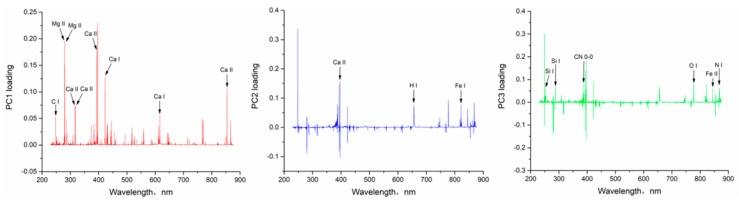
The loading plots of PC1, PC2, and PC3.

**Table 1 sensors-19-01453-t001:** The results of the extreme learning machine (ELM), the soft independent modeling of class analogy (SIMCA), K-nearest neighbor (KNN) and random forest (RF) models based on LIBS full spectrum.

Discriminant Analysis Model	Parameter ^[a]^	Accuracy of Calibration Set	Accuracy of Prediction Set
ELM	53	100%	99.30%
SIMCA	(7,12,11,8,2)	100%	99.00%
KNN	3	100%	100%
RF	(151,7)	100%	100%

^[a]^ The number of neurons for ELM, the number of principal components (PCs) of SIMCA, the *k* value for KNN, and number of trees in the forest and nodes per tree for RF.

**Table 2 sensors-19-01453-t002:** Eighteen characteristic wavelengths corresponding to the elements.

Number	Wavelength (nm)	Element	Number	Wavelength (nm)	Element
1	247.86	C I	10	396.85	Ca II
2	251.61	Si I	11	422.67	Ca I
3	279.55	Mg II	12	616.38	Ca I
4	280.27	Mg II	13	656.28	H I
5	288.15	Si I	14	777.54	O I
6	315.89	Ca II	15	821.58	Fe I
7	317.93	Ca II	16	844.80	Fe II
8	388.22	CN 0-0	17	854.21	Ca II
9	393.37	Ca II	18	868.02	N I

**Table 3 sensors-19-01453-t003:** Results of discriminant analysis model based on characteristic wavelength.

Discriminant Analysis Model	Parameter	Accuracy of Calibration Set	Accuracy of Prediction Set
ELM	118	100%	99.30%
SIMCA	(3,4,3,3,2)	100%	98.00%
KNN	3	100%	100%
RF	(151,7)	100%	100%

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
