# Peer review of "Rapid Identification of Kudzu Powder of Different Origins Using Laser-Induced Breakdown Spectroscopy"

_sensors, 2019, doi:10.3390/s19061453_

Round 1
Reviewer 1 Report
The authors present experimental detection of chinese medicine (kudzu) powders from five different provinces in China using laser induced breakdown spectroscopy (LIBS). They claim that the combination of LIBS and statistical analysis can be used to distinguish the different origins and quality of the samples in a minimally invasive and rapid way. The results are interesting and potentially useful. The manuscript can be accepted after the following issues are addressed:
1) The title is confusing. The authors mention "rapid identification" but they did not discuss what means "rapid." They should provide the details of the time of experiment that it takes to prepare the sample and to record, analyze the data and to use the statistical tools. It seems that it will not be rapid enough but may have a potential to become rapid in the future by making an automated device.
2) The authors should provide more experimental details. For example, laser pulse duration, focusing lens focal distance, laser beam diameter at the sample.
3) The authors should describe in more detail how they selected 18 wavelengths for LIBS analysis and labels those 18 peaks in the PCA plots in figure 4.
4) The authors should discuss the origins of the differences in the samples and LIBS signals. LIBS shows element concentrations such as Fe, Ca, etc. The authors should discuss why some samples contain different amounts of these elements and if possible perform control experiments.
5) English should be improved.
Author Response
Response to Reviewer 1 Comments
Point 1: The title is confusing. The authors mention "rapid identification" but they did not discuss what means "rapid." They should provide the details of the time of experiment that it takes to prepare the sample and to record, analyze the data and to use the statistical tools. It seems that it will not be rapid enough but may have a potential to become rapid in the future by making an automated device.
Response 1: Thank you for your suggestion. The "rapid identification" in the title is proposed in relation to the problem that the traditional detection methods such as AAS, AFS and ICP-MS take a long time. The whole sample pretreatment for acquisition of LIBS signals was less than 2 min including weighing and pressing, while the pretreatment for the other common analytical techniques such as AAS, AFS and ICP-MS needs more than 150 min and contains weighing, adding other reagent, digesting, discharging acid and diluting, etc. After sample pretreatment, time of LIBS information collection and data analysis for one sample is about 2 min and is compatible with the requirements of on-site analysis. The explanation in the manuscript may not clear, we have corrected the sentence as “The whole sample pretreatment for acquisition of LIBS signals was less than 2 min including weighing and pressing.” in Line 129-130 and “The time of LIBS information collection for one sample was about 1 min. And it’s compatible with the requirements of on-site analysis.” in Line 146-147.
Point 2: The authors should provide more experimental details. For example, laser pulse duration, focusing lens focal distance, laser beam diameter at the sample.
Response 2: Sorry, experimental details is not clear and we have made supplements including laser pulse duration, focusing lens focal distance and laser beam diameter in Line 103-107 in the revised manuscript: “Based on previous research, Q-switched Nd:YAG pulsed laser (Vlite-200, Beamtech Optronics, Beijing China) was used to generate pulse with the maximal energy of 200 mJ @532 nm, pulse duration of 8 ns. The laser was emitted and passed through the optical system. Finally, under the action of a plano-convex lens (f = 100 mm), the laser was focused 2 mm below the sample and formed a spot with a diameter of 7 mm.”
Point 3: The authors should describe in more detail how they selected 18 wavelengths for LIBS analysis and labels those 18 peaks in the PCA plots in figure 4.
Response 3: Thank you for your question. The more details how they selected 18 wavelengths for LIBS analysis are described in Line 269-277 in the revised manuscript: “In Fig. 4, the loading values of variables at the positive or negative peaks had larger absolute values and were more important to the full data. When processing the data, we used 0.05 as the reference value and selected the variable, whose absolute value of the loading was greater than 0.05, as the alternative key variable of the principal component. Using the above method, alternative key variables for PC1, PC2 and PC3 could be selected according to Fig. 4. Then then intensity of spectral line and the distance between the spectral peak and other surrounding peaks of the above alternative key variables were comprehensively compared. The 18 spectral lines at the peak or trough were selected as the key variables in Fig. 4, based on the above selection conditions of characteristic spectral lines.” Those 18 peaks in the PCA plots are labelled in Fig. 4.
Figure 4. The loading plots of PC1, PC2, PC3.
Point 4: The authors should discuss the origins of the differences in the samples and LIBS signals. LIBS shows element concentrations such as Fe, Ca, etc. The authors should discuss why some samples contain different amounts of these elements and if possible perform control experiments.
Response 4: Thank you for your question. The origins of the differences in the samples and LIBS signals have been added in Line 215-218 in the revised manuscript: “It was found that the intensity of Mg II (279.55 nm, 280.27 nm), Ca II (393.37 nm, 396.85 nm) and Ca I (422.67 nm) of kudzu powder from Yunnan is much stronger than 7.0×105. But the peak of these elements in the other four samples was much lower than it.” The different samples come from five different origins. Different regions have different ecological environments such as soil, temperature, water, light and altitude. The environmental conditions directly affect the quality of samples, including active ingredients (mineral nutrition element), appearance traits (color and size), and hazardous substances (pesticide residues and heavy metals). Therefore, for samples from different regions, there may be differences in the content of the same elements. It is the reason why some samples contain different amounts of these elements.
Point 5: English should be improved.
Response 5: Thank you for your suggestion. Some expressions have been modified in revised manuscript. And I will pay more attention to improving English in the future.

Reviewer 2 Report
The manuscript entitled ‘Rapid identification of kudzu powder of different origin using laser-induced breakdown spectroscopy’ is understandable and provides useful information on the identification of kudzu powder of different origin. However, I do have some comments on the manuscript that requires revision in my opinion.
2.1 Section: Experimental Setup
1) Please provide more information about the spectrometer used. What spectral range does it cover and what was the resolution of the wavenumbers.
2.3 Section: Data acquisition
2) Please clarify when using the X-Y-Z movable sample stage, is that motorized? Also, what’s the distance between locations in the sample, are they evenly distributed by different rows and columns? Please specify how those locations were chosen.
3) Each point was cumulatively acquired 10 times, please specify if a single spectrum was obtained per location (cumulative of the 10 shots) or 10 separate spectra were obtained per location (non cumulative). As an average of 160 spectral data was taken I assume these were non cumulative? Please clarify
4) The term LIBS is used throughout the whole manuscript except in this section where LIPS is used, please correct.
5) Please provide further information on sample analysis, were all 250 samples measured on the same day? Was a particular order of acquisition used or were samples randomly analysed? Ideally samples should be analysed in a random order and in different independent occasions to ensure the spectral data acquired is not influenced over time. Intensity strengths could change over time and give the false assumption that samples are different.
6) As multiple shots are taken per location please specify the speed in Hz between shots used.
7) What was the timeframe between sample preparation and LIBS analysis, were samples analysed on the same day of producing the pellets or were they stored for days?
2.5 Section: Discriminant analysis method
8) Please check the spelling for SLFNN (neural networks)
3.1 Section: Spectral analysis of different samples
9) Please justify the assignation of the spectral lines by referencing the atomic spectral database of NIST.
10) Figure 2: Please specify is the spectra presented the result of one random shot of a random sample or is it the average spectra of all replicates per sample origin. Ideally the average spectrum should be presented to ensure it is representative of each sample origin, especially as the main difference observed is based on different intensities
11) “It has been shown that there was a linear relationship between the peak intensity of the element in the laser-induced breakdown spectrum and the element content. “ Please justify this sentence, what is the element content of the samples? Provide a table or reference if necessary
3.2 Section: PCA
12) The scatter plots shows one sample of Yunam that seems to be an outlier, why is that? if this is the case, that sample should not be used to build the calibration models
13) Two subgroups of the same sample origin can clearly be seen for Zheijiang and Hubei samples, is there an explanation for this behaviour? Were samples analysed at different times?
3.3 Section: Discriminant analysis model based on full spectrum
14) Please check the spelling for SIMCA model throughout the text
3.4 Section: Discriminant analysis model based on characteristic wavelength
15) Explain what’s your criteria for selecting the 18 spectral lines chosen, the PCA clearly shows more lines than 18, if it is based on the loading value, what value is this?
Author Response
Response to Reviewer 2 Comments
2.1 Section: Experimental Setup
Point 1: Please provide more information about the spectrometer used. What spectral range does it cover and what was the resolution of the wavenumbers.
Response 1: Thank you for your question. The information about the spectrometer used has been added in Line 109 in the revised manuscript: “The spectra between 200–975 nm with high resolution (λ/Δλ = 5000) was collected.”
2.3 Section: Data acquisition
Point 2: Please clarify when using the X-Y-Z movable sample stage, is that motorized? Also, what’s the distance between locations in the sample, are they evenly distributed by different rows and columns? Please specify how those locations were chosen.
Response 2: Thank you for your reminder. The X-Y-Z movable sample stage is motorized. Spectral data acquisition of the sample is performed on 16 locations. The locations in the sample are evenly distributed and the distance between each row and each column is 2 mm. The details about data acquisition and locations selection have been added in Line 138-147 in the revised manuscript: “Considering the impacts caused by internal component differences and surface impurities of samples, the following methods were used for data acquisition. In order to avoid continuous ablation of the same spot, 4×4 array was set as the ablation path by X-Y-Z motorized stage, where the distance between each row and each column is 2 mm. As a result, spectral data acquisition of the sample was performed on 16 locations, which are evenly distributed on each sample. To reduce fluctuation between the laser point-to-point and get a stable signal, each point was cumulatively acquired 5 times with repetition rate of 1 Hz. Finally, the average of the 80 spectra (4×4×5) was recorded as the LIBS data of the sample. A total of 250 LIBS spectral data represented 250 samples of kudzu powder were obtained in the experiment. The time of LIBS information collection for one sample was about 1 min. And it’s compatible with the requirements of on-site analysis.”
Point 3: Each point was cumulatively acquired 10 times, please specify if a single spectrum was obtained per location (cumulative of the 10 shots) or 10 separate spectra were obtained per location (non cumulative). As an average of 160 spectral data was taken I assume these were non cumulative? Please clarify.
Response 3: Firstly, we are sorry to say that due to our mistakes, the cumulative number of each point should be 5, which we mistakenly wrote as 10 in the previous manuscript. The cumulative number of each point has been added in Line 142-144 in the revised manuscript: “To reduce fluctuation between the laser point-to-point and get stable signals, each point was cumulatively acquired 5 times with repetition rate of 1 Hz.” In the experiment, each point was cumulatively acquired 5 times, a single spectrum was obtained per sample (cumulative of the 80 shots).
Point 4: The term LIBS is used throughout the whole manuscript except in this section where LIPS is used, please correct.
Response 4: Sorry, the spelling mistake of LIBS has been corrected in the revised manuscript.
Point 5: Please provide further information on sample analysis, were all 250 samples measured on the same day? Was a particular order of acquisition used or were samples randomly analysed? Ideally samples should be analysed in a random order and in different independent occasions to ensure the spectral data acquired is not influenced over time. Intensity strengths could change over time and give the false assumption that samples are different.
Response 5: Thank you for your question. Due to the existence of the matrix effect, the LIBS technology is more susceptible to the experimental environment such as temperature and moisture. Therefore, in the experiment, the measurement of all 250 samples was completed on the same day. In the experiment, five kinds of samples from different origins were selected. For the same sample, the LIBS spectra were collected in a random order.
Point 6: As multiple shots are taken per location please specify the speed in Hz between shots used.
Response 6: Thank you for your reminder. The repetition rate of Q-switched Nd:YAG pulsed is laser from 1 to 10 Hz. In the experiment, the speed between shots is 1 Hz. We have added this aspect in Line 142-144 in revised manuscript: “To reduce fluctuation between the laser point-to-point and get stable signals, each point was cumulatively acquired 5 times with repetition rate of 1 Hz.”
Point 7: What was the timeframe between sample preparation and LIBS analysis, were samples analysed on the same day of producing the pellets or were they stored for days?
Response 7: Thank you for your question. The timeframe between sample preparation and LIBS analysis is one day. The LIBS analysis was performed the next day after pretreatment of all samples. Before the LIBS experiment, all samples are stored in the sample cabinet with constant temperature and humidity to prevent sample quality changes.
2.5 Section: Discriminant analysis method
Point 8: Please check the spelling for SLFNN (neural networks)
Response 8: Thank you for your reminder. The spelling mistakes of SLFNN (neural networks) have been corrected in the revised manuscript.
3.1 Section: Spectral analysis of different samples
Point 9: Please justify the assignation of the spectral lines by referencing the atomic spectral database of NIST.
Response 9: Thank you for your reminder. The assignation of the spectral lines was justified by the National Institute of Standards and Technology (NIST) Atomic Spectra Database (ASD). But we didn’t mentions it in the previous manuscript. This aspect has been added in Line 195-197 in revised manuscript: “Based on the National Institute of Standards and Technology (NIST) Atomic Spectra Database (ASD), some strong emission lines were observed in all five kinds of kudzu powder.”
Point 10: Figure 2: Please specify is the spectra presented the result of one random shot of a random sample or is it the average spectra of all replicates per sample origin. Ideally the average spectrum should be presented to ensure it is representative of each sample origin, especially as the main difference observed is based on different intensities
Response 10: Thank you for your reminder. The spectra presented in Fig.2 is the average spectra of Kudzu powder samples. The name of Fig.2 has been changed in Line 203-204 in revised manuscript: “Figure 2. LIBS Average original spectraum of Kudzu powder samples. (a)Zhejiang; (b)Hubei; (c)Anhui; (d)Hunan; (e)Yunnan.”
Point 11: “It has been shown that there was a linear relationship between the peak intensity of the element in the laser-induced breakdown spectrum and the element content. “ Please justify this sentence, what is the element content of the samples? Provide a table or reference if necessary
Response 11: Thank you for your reminder. It is not clear from the Fig.2 whether there is obvious linear relationship between the peak intensity of the element in the laser-induced breakdown spectrum and the element content. Sorry. It’s our unclear expression that made the reviewer misunderstood. What we originally intended to explain is as follows. Previous studies have shown that there is a linear relationship between the peak intensity of the element in the LIBS signal and the content of the elements in samples. It provides a way for us to preliminarily judge and compare the content of elements by observing and analyzing the peak intensity of elements in the spectrums.
3.2 Section: PCA
Point 12: The scatter plots shows one sample of Yunnan that seems to be an outlier, why is that? if this is the case, that sample should not be used to build the calibration models
Response 12: Thank you for your reminder. As the reviewer comments, one sample of Yunnan is an outlier. This outlier was removed before building the calibration models. Sorry. Fig.3 was not updated due to negligence. Fig.3 has been changed in Line 236 in revised manuscrip.
Figure 3. Scores scatter plots of PC1, PC2 and PC3 of different samples. (a) Plots of PC1-PC2; (b) Plots of PC2-PC3; (c) Plots of PC1-PC2-PC3.
Point 13: Two subgroups of the same sample origin can clearly be seen for Zheijiang and Hubei samples, is there an explanation for this behaviour? Were samples analysed at different times?
Response 13: Thank you for your question. In experiments, the samples were analysed at on the same day. The reason for this behaviour may be that the samples are of different batches. It can be seen from Fig.3 that there are five distinct clusters and five kinds of samples can be distinguished from other kinds. For samples of Anhui, Hunan and Yunnan, there is no subgroup of the same sample and the same sample can be well clustered together. For Zhejiang and Hubei samples, although they have two obvious subgroups of the same sample, they are obviously different from other samples and can be clearly distinguished from other kinds of samples. The paper is aim to study the feasibility of rapidly identifying kudzu powder from different producing areas. In future studies, we will research the identification and classification from different batches of same samples and discuss the effect of experiment time on LIBS.
3.3 Section: Discriminant analysis model based on full spectrum
Point 14: Please check the spelling for SIMCA model throughout the text.
Response 14: Thank you for your reminder. The spelling mistakes of SIMCA model have been corrected in the revised manuscript.
Section: Discriminant analysis model based on characteristic wavelength
Point 15: Explain what’s your criteria for selecting the 18 spectral lines chosen, the PCA clearly shows more lines than 18, if it is based on the loading value, what value is this?
Response 15: Thank you for your question. The criteria for selecting the 18 spectral lines is explained as follows: In the loading plots of principal component, the larger the absolute value of the loading values corresponding to the variables, which are at the positive or negative peak, the higher the importance to the full spectrum data. In Fig. 4, the loading values of variables at the positive or negative peaks had larger absolute values and were more important to the full data. When processing the data, we used 0.05 as the reference value and selected the variable, whose absolute value of the loading was greater than 0.05, as the alternative key variable of the principal component. Using the above method, alternative key variables for PC1, PC2 and PC3 could be selected according to Fig. 4. Then then intensity of spectral line and the distance between the spectral peak and other surrounding peaks of the above alternative key variables were comprehensively compared. The 18 spectral lines at the peak or trough were selected as the key variables in Fig. 4, based on the above selection conditions of characteristic spectral lines.